# Comparative Analysis of Aerotolerance, Antibiotic Resistance, and Virulence Gene Prevalence in *Campylobacter jejuni* Isolates from Retail Raw Chicken and Duck Meat in South Korea

**DOI:** 10.3390/microorganisms7100433

**Published:** 2019-10-10

**Authors:** Jinshil Kim, Hyeeun Park, Junhyung Kim, Jong Hyun Kim, Jae In Jung, Seongbeom Cho, Sangryeol Ryu, Byeonghwa Jeon

**Affiliations:** 1Department of Food and Animal Biotechnology, Research Institute for Agriculture and Life Sciences, and Center for Food and Bioconvergence, Seoul National University, Seoul 08826, Korea; jinsilk1130@naver.com (J.K.); bluejoka@naver.com (H.P.); 2Department of Agricultural Biotechnology, Seoul National University, Seoul 08826, Korea; 3College of Veterinary Medicine and Research Institute for Veterinary Science, Seoul National University, Seoul 08826, Korea; tpkimjh@snu.ac.kr (J.K.); chose@snu.ac.kr (S.C.); 4The Korea Centers for Disease Control and Prevention, Osong 28159, Korea; micro487@hanmail.net (J.H.K.); jeon_bh@hotmail.com (J.I.J.); 5Environmental Health Sciences, School of Public Health, University of Minnesota, Minneapolis, MN 55455, USA

**Keywords:** *Campylobacter jejuni*, retail poultry, aerotolerance, antibiotic resistance, virulence gene prevalence

## Abstract

Human infections with *Campylobacter* are primarily associated with the consumption of contaminated poultry meat. In this study, we isolated *Campylobacter*
*jejuni* from retail raw chicken and duck meat in Korea and compared their aerotolerance, antibiotic resistance, and virulence gene prevalence. Whereas *C. jejuni* isolates from chicken dominantly belonged to multilocus sequence typing (MLST) clonal complex (CC)-21, CC-45 is the common MLST sequence type in duck meat isolates. *C. jejuni* strains from both chicken and duck meat were highly tolerant to aerobic stress. The prevalence of virulence genes was higher in *C. jejuni* strains from chicken than those from duck meat. However, antibiotic resistance was higher in duck meat isolates than chicken isolates. Based on the prevalence of virulence genes and antibiotic resistance, fluoroquinolone-resistant *C. jejuni* strains harboring all tested virulence genes except *vir*B11 were predominant on retail poultry. Fluoroquinolone-resistant *C. jejuni* strains carrying most virulence genes were more frequently isolated in summer than in winter. The comparative profiling analysis in this study successfully demonstrated that antibiotic-resistant and pathogenic strains of *C. jejuni* are highly prevalent on retail poultry and that retail duck meat is an important vehicle potentially transmitting *C. jejuni* to humans in Korea.

## 1. Introduction

*Campylobacter* is a leading etiological agent of gastroenteritis, accounting for approximately 166 million diarrheal cases globally per year [1]. Although estimates of the number of human campylobacterisosis cases in South Korea are not available, *Campylobacter* is considered as a major food safety concern in the country along with enterotoxigenic *Escherichia coli*, *Salmonella*, and *Clostridium* according to the Ministry of Food and Drug Safety (MFDS). (https://www.foodsafety korea.go.kr/portal/healthyfoodlife/foodPoisoningStat.do)

*Campylobacter jejuni* is the pathogenic species most frequently implicated in human infection and may result in diarrhea, fever and abdominal pains. In some cases, *C. jejuni* may lead to the onset of Guillain–Barré syndrome, an autoimmune disorder characterized by acute and progressive neuromuscular paralysis [2]. 

The pathogenicity of *C. jejuni* is mediated by various virulence factors, such as flagellin, cytolethal distending toxin (CDT), adhesins (e.g., *Campylobacter* adhesion to Fibronectin (CadF), Peb1), *Campylobacter* invasion antigen (Cia) proteins, and phospholipase A (PldA). Flagellins are the building block of flagella that play an important role in motility, chicken colonization, autoagglutination, and biofilm formation in *Campylobacter* [3]. The CDT toxin consists of three subunits (CdtA, CdtB, and CdtC) [4], and CdtB possesses a type I deoxyribonuclease activity and arrests the cell cycle in the G_2_/M transition phase [5]. Peb1 is an adhesin that is involved in *C. jejuni* interaction with INT 407 cells [6] and affects host colonization [7]. CadF is an outer membrane protein that affects *C. jejuni* binding to epithelial cells [8], and a mutation of *cad*F completely prevented *C. jejuni* from colonizing chicken intestines [9]. CiaB is associated with the internalization of *C. jejuni* into host cells [10]. PldA is an outer-membrane protein with a hemolytic activity [11]. PldA and CiaB also significantly affect the ability of *C. jejuni* to colonize chicken intestines [12].

Human campylobacteriosis is usually self-limiting. However, severe and prolonged infections may require antimicrobial therapy [13], and fluoroquinolone (FQ) and macrolide antibiotics are the drugs of choice to treat human campylobacteriosis [14,15]. For serious *Campylobacter* infections in humans, such as bacteremia, aminoglycosides are intravenously administered to patients [16]. However, increasing resistance in *Campylobacter* to clinically important antibiotics, particularly FQs, is a serious global public health problem, and the World Health Organization (WHO) recently classified FQ-resistant *Campylobacter* as a high-priority antibiotic-resistant pathogen for which new drugs need to be developed [17].

*C. jejuni* is a microaerophilic bacterium and requires low oxygen concentrations for growth [18]. Thus, tolerance to aerobic stress is the forefront survival mechanism for *C. jejuni* in the food production and processing systems in a normal atmosphere where the oxygen level is high (ca. 21%). In contrast to our traditional knowledge about oxygen sensitivity in *C. jejuni*, recently, *C. jejuni* strains with increased aerotolerance have been isolated from retail raw chicken and humans [19,20]. In Canada, *C. jejuni* strains with increased aerotolerance are highly prevalent in retail raw chicken [20]. Additionally, aerotolerant *C. jejuni* survived on raw chicken under aerobic conditions for an extended period of time compared to oxygen-sensitive *C. jejuni* strains, potentially posing a serious threat to food safety [21].

*Campylobacter* is isolated from a wide range of food-producing animals and their carcasses [22], and poultry is considered as the primary reservoir for *Campylobacter* [23]. It has been estimated that 50% to 80% of human cases of campylobacteriosis are attributed to chickens, and 20% to 30% are directly associated with the handling, preparation, and consumption of broiler meat in the EU [24]. In addition, *Campylobacter* is isolated from duck meat and its parts in many countries [25], and the consumption of duck meat and organs (e.g., liver pâté) has caused *Campylobacter* outbreaks [26,27]. While turkey is widely consumed in North America, duck is preferably consumed in most Asian countries including Korea. Since duck meat has been increasingly produced and consumed in Korea, human exposure to *Campylobacter* can be caused by eating contaminated duck meat. However, little has been investigated about the food safety risk of *C. jejuni* on duck meat.

To fill this knowledge gap, in this study, we isolated *C. jejuni* from retail chicken and duck meat in Korea and analyzed their multilocus sequence typing (MLST) clonal complexes (CCs), aerotolerance, virulence gene prevalence, and antibiotic resistance.

## 2. Materials and Methods

### 2.1. Isolation of C. jejuni from Retail Raw Chicken and Duck Meats

Eighty poultry whole carcasses (52 chickens and 28 duck meats) were collected from retail stores in seven different provinces in Korea in winter (from December 2016 to March 2017), and 114 poultry meats (81 chickens and 33 ducks) were purchased in summer (from April to June 2017). A total of 194 whole carcasses were subjected to enrichment with 1 L of Bolton broth supplemented with Bolton *Campylobacter*-selective supplement (Oxoid, UK) under microaerobic conditions (4% H_2_, 6% O_2_, 7% CO_2_, 83% N_2_) at 42 °C for 24 h. To increase isolation efficiency, an aliquot (20 mL) of the enriched broth was concentrated and resuspended in 1 mL of Bolton broth. The resuspension was serially diluted (10^0^ to 10^−5^) and spread on six Preston agars supplemented with Preston *Campylobacter*-selective supplement (Thermo-Fisher Scientific, Waltham, MA, USA). The culture was incubated at 42 °C for 48 h under microaerobic conditions. 

### 2.2. Identification of Campylobacter Species 

Presumptive *Campylobacter* colonies were selected based on the typical colony morphology of *Campylobacter*, such as flat, shiny, and mucoid colonies. To avoid analyzing duplicate clones, only 1~2 isolates were selected from each sample depending on different colony morphologies for further confirmation using multiplex PCR and 16S rRNA sequencing. Since *hipO* has been used as a specific marker for *C. jejuni* in many studies, the isolates that were positive for *hipO* and *cj0414* were considered as *C. jejuni*, and those that were positive for *ask* were determined as *Campylobacter coli* [28]. To prepared template DNA, the pure culture in Muller Hinton (MH, Oxoid) broth resuspended in MH broth to an optical density at 600 nm (OD_600_) of 0.1 was diluted 10-fold in distilled water and heated to 95 °C for 7 min to extract DNA. Cell debris was pelleted by centrifugation at 16,000 × *g*, 4 °C for 1 min, and the supernatant was used as template DNA. The isolates were further differentiated with 16S rRNA sequencing. The primer sequences were described in Appendix A.

### 2.3. MLST Analysis

*C. jejuni* genotypes were characterized by MLST based on the method outlined in PubMLST (pubmlst.org) and a previous study [29]. Briefly, template DNA was prepared as described above. The seven housekeeping genes (*asp*A, *gln*A, *glt*A, *gly*A, *pgm*, *tkt*, and *unc*A) were amplified by PCR with the primer pairs described elsewhere [29]. The PCR amplicons were commercially sequenced by Macrogen (Seoul, Korea). Allele numbers and sequence types were assigned by using the *Campylobacter* MLST database [30].

### 2.4. Aerotolerance Test

An aerotolerance test was performed as described previously [20]. Briefly, overnight cultures of *C. jejuni* on MH agar at 42 °C were resuspended in MH broth to an OD_600_ of 0.1. The bacterial suspension was incubated at 42 °C with shaking (200 rpm) under aerobic conditions. Samples were taken after 0, 12, and 24 h for serial dilution and bacterial counting. The strains that survived in aerobic shaking < 12 h were called oxygen-sensitive (OS), and those that survived 12~24 h and ≥ 24 h were considered aerotolerant (AT) and hyper-aerotolerant (HAT), respectively, according to the previous study [20].

### 2.5. PCR Detection of Virulence Genes

The prevalence of nine virulence genes in *C. jejuni* was examined with single PCR. Template DNA was prepared as described above. The PCR reaction was performed with rTaq (Takara, Japan) under the following conditions: initial denaturation at 95 °C for 15 min followed by 35 cycles of denaturation at 95 °C for 30 s, variable annealing temperature (*cad*F, *cia*B, *iam*, *pld*A, and *fla*A at 45 °C, *doc*A, and *peb*1 at 50 °C, *vir*B11 at 53 °C, and *cdt*B at 58 °C) for 1 min 30 s, extension at 72 °C for 2 min and the final extension at 72 °C for 7 min. The results were analyzed by electrophoresis with 1% agarose gels. The primer sequences are described in Appendix A. 

### 2.6. Antimicrobial Susceptibility Test

The minimum inhibitory concentrations (MICs) of the *C. jejuni* isolates were determined using a broth microdilution method with the Sensititre custom plate KRCAMP (TREK Diagnostics, Cleveland, OH, USA). The antimicrobials in the plate were two-fold diluted, including azithromycin, chloramphenicol, ciprofloxacin, erythromycin, gentamicin, nalidixic acid, streptomycin, tetracycline, and telithromycin. *C. jejuni* was suspended in cation-adjusted MH broth to a 0.5 McFarland standard. The microtiter plates were incubated at 42 °C for 24 h under microaerophilic conditions. Antimicrobial resistance was determined according to the interpretative criteria recommended by the Clinical and Laboratory Standards Institute [31], and a report from the US Food and Drug Administration [32]. *C. jejuni* ATCC 33560 was included in the test as a quality control strain. 

### 2.7. Data Analysis

The data on virulence genes and the MIC test were analyzed using BioNumerics 7.6.2 (Applied Maths, Belgium). Cluster analysis was performed, and a dendrogram was generated with BioNumerics using the similarity coefficient and unweighted-pair group method with average linkages (UPGMA) coefficient. 

### 2.8. Statistical Analysis

Statistical analyses were performed using χ^2^ test with SPSS (IBM, USA).

## 3. Results

### 3.1. Frequencies of Campylobacter Isolation from Retail Raw Chicken and Duck Meat 

*Campylobacter* was isolated from 54.1% (105/194) of the total poultry meat samples and more frequently from duck (62.3%) than chicken (50.4%) meat (Table 1). Compared to *C. coli*, *C. jejuni* was more prevalent on retail raw poultry in Korea, and both pathogenic species were simultaneously isolated from 9.8% of raw chicken and 18.0% of duck meat samples (Table 1). The isolation frequencies of *Campylobacter* from raw chicken were higher in summer than in winter; however, it was not statistically significant (*p* = 0.0648; Table 1). However, such a seasonal pattern was not observed in duck meat samples (Table 1). These results showed that *Campylobacter* contamination was highly prevalent on both raw chicken and duck meat and that retail raw duck meat is highly prone to *Campylobacter* contamination. The rest of the study was focused on the characterization of *C. jejuni* isolates.

### 3.2. MLST Analysis of C. jejuni Isolates from Raw Retail Chicken and Duck Meat

One hundred and thirty five strains of *C. jejuni* were isolated from 105 *C. jejuni*-positive samples (64 chicken and 41 duck meat samples); we selected only one or two isolates from each sample based on their morphological differences to avoid isolating duplicate clones. The results of the MLST analysis showed that *C. jejuni* strains belonging to CC-21 and CC-45 were dominant on both raw chicken and duck meats (Figure 1A). Whereas CC-21 was predominant in chicken isolates, CC-45 was highly prevalent in *C. jejuni* isolates from duck meat (*p* < 0.05; Figure 1A). The MLST CCs of 28.9% of duck meat isolates were not assigned (*p* < 0.05), and 17.8% of them were not typable (*p* < 0.05) with MLST; however, 14.4% of the chicken isolates were not assigned with MLST CC or non-typable (*p* < 0.05; Figure 1A). This suggests that the genetic background of certain *C. jejuni* isolates from duck meat may be different from that of chicken isolates.

### 3.3. Aerotolerance in C. jejuni Isolates from Raw Chicken and Duck Meat

We investigated aerotolerance in *C. jejuni* isolates from raw chicken and duck meat. AT and HAT strains of *C. jejuni* were highly prevalent in both raw chicken and duck meat (Table 2). Although the prevalence of HAT *C. jejuni* was higher in raw chicken than duck meat, the combined prevalence of AT and HAT *C. jejuni* strains was similar between chicken isolates and duck meat isolates (86.7% in chicken isolates vs 86.6% in duck meat isolates; Table 2). MLST CC-21 was most common in AT and HAT strains of *C. jejuni* from retail raw chicken (Figure 1B). In addition, most (66.7%) OS strains from duck meat were non-typable (*p* < 0.05); however, CC-45 was predominant in AT and HAT strains (*p* < 0.05) of *C. jejuni* from duck meat (Figure 1B). The results suggest that the genetic background may differ in OS, AT and HAT strains of *C. jejuni* from retail poultry. 

### 3.4. Antibiotic Resistance in C. jejuni Isolates from Retail Raw Chicken and Duck Meat 

*C. jejuni* strains isolated from retail raw chicken and duck meat were highly resistant to ciprofloxacin and tetracycline (Table 3 and Appendix A). FQ resistance was higher in duck meat isolates (97.8%) than chicken isolates (*p* < 0.05; 85.6%) (Table 3). Similarly, higher rates of tetracycline resistance were observed in *C. jejuni* isolates from duck meat (57.8%) than chicken (*p* < 0.001; 27.8%) (Table 3). In contrast to the high levels of FQ and tetracycline resistance, only a single strain of *C. jejuni* was resistant to macrolides (e.g., erythromycin and azithromycin), the most important antibiotic class of clinical importance for treating human campylobacteriosis, and five strains were resistant to gentamicin (Table 3). The findings demonstrated that *C. jejuni* isolates from duck meat are highly resistant to antibiotics, particularly FQs and tetracycline. 

### 3.5. Virulence Gene Prevalence and Antibiotic Resistance in C. jejuni Isolates from Retail Raw Chicken and Duck Meat 

To evaluate the virulence potential of *C. jejuni* isolates, the prevalence of virulence genes was determined with PCR. A total of 68.9% (93/135) of *C. jejuni* isolates harbored all tested virulence genes except *vir*B11 (Appendix A), which is a virulence gene encoded on the plasmid pVir [33]. Approximately 74.4% (67/90) of *C. jejuni* strains from raw chicken harbored eight virulence genes (all except *vir*B11), and 7.8% (7/90) of chicken isolates possessed all tested virulence genes (Appendix A), indicating that 82.2% (74/90) of chicken isolates carried at least eight virulence genes involved in colonization, adhesion, and invasion (Appendix A). Among 45 duck isolates, 57.8% (26/45) possessed eight virulence genes (all except *vir*B11), and 2.2% (1/45) of duck strains harbored all tested virulence genes (Appendix A), meaning that 60.0% (27/45) of duck meat isolates carried more than eight virulence genes. The prevalence of virulence genes was higher in *C. jejuni* strains from chicken compared to duck meat strains (Table 4). The virulence genes were widely distributed in *C. jejuni* isolates from chicken regardless of the level of aerotolerance. Among duck meat isolates of *C. jejuni*, the prevalence of *cad*F, *iam*, *pld*A, *doc*A, *peb*1, and *fla*A was higher in AT and HAT strains compared to OS strains (Appendix A). 

### 3.6. Integrative Comparison of Virulence Gene Prevalence and Antibiotic Resistance between C. jejuni Chicken Isolates and Duck Meat Isolates

Since both virulence factors and antibiotic resistance affect food safety associated with *Campylobacter*, 135 *C. jejuni* strains from retail raw chicken and duck meats were analyzed based on virulence gene prevalence and antibiotic resistance. In this study, we named it the virulence and antibiotic resistance (Viro and AMR) type. The analysis clustered the 135 *C. jejuni* isolates from chicken and duck meat into 30 different Viro and AMR types. Types 4 and 13 constituted 36.3% (49/135) and 20.0% (27/135) of the total strains, respectively (Figure 2A). *C. jejuni* strains in Type 4 harbored all tested virulence genes except *vir*B11 and were resistant to FQs. Type 4 was predominant (42.2%; 38/90) with a statistical significance (*p* < 0.05) in chicken isolates, whereas Type 13 was highly prevalent (28.9%; 13/45) in duck meat isolates (Figure 2A). Type 13 was highly dominant and exhibited the same patterns of virulence gene prevalence and antibiotic resistance as Type 4 except for tetracycline resistance (Figure 2A). *C. jejuni* strains in Types 1 and 2, which harbored all tested virulence genes, were primarily isolated from chicken (Figure 2A). 

*C. jejuni* strains belonging to Type 4 were prevalent in AT and HAT strains. The proportion of Type 4 was 27.8%, 35.6%, and 40.9% in OS, AT, and HAT strains, respectively (Figure 2B and Appendix A). In contrast, Type 13 was less prevalent in HAT strains, constituting 27.8%, 19.2%, and 18.2% in OS, AT, and HAT strains, respectively (Figure 2B and Appendix A). Whereas Type 4 was predominant in chicken isolates, Type 13 was predominant in duck meat isolates (Figure 2B). Furthermore, *C. jejuni* strains belonging to Types 4 and 13 were isolated more frequently in summer than in winter (Figure 2B). 

*C. jejuni* strains in Types 4 and 13 exhibited different compositions of MLST sequence types. Whereas CC-21 was predominant in *C. jejuni* strains in Type 4 from both chicken and duck meats (*p* < 0.05; Figure 3), CC-45 was the major sequence type in Type 13 in both chicken and duck meat strains (*p* < 0.001; Figure 3). In addition, all nine *C. jejuni* strains (six from raw chicken and three from duck meats) belonging to CC-443 (Figure 1A) were clustered to Type 4 (Figure 3). The integrative analysis of virulence gene prevalence and antibiotic resistance suggest that antibiotic-resistant *C. jejuni* strains harboring virulence genes are highly prevalent on retail duck meat. 

## 4. Discussion

Aerotolerance in *Campylobacter* may directly affect food safety, as it increases the survival of this microaerophilic pathogen in the food supply chain. In this study, we first reported aerotolerance in *C. jejuni* strains from duck meat. Similar to a previous study in Canada [20], the AT *C. jejuni* strains is highly prevalent on both retail raw chicken and duck meat in Korea. As a foodborne pathogen, *C. jejuni* encounters various stress conditions at different stages of food production, processing, preservation, distribution, and cooking, and aerobic stress would be a common stressor to this microaerophilic bacterium in the normal atmosphere. As it has been demonstrated that AT and HAT strains of *C. jejuni* survives on refrigerated raw chicken in air more effectively than OS strains [21], the high prevalence of AT and HAT strains of *C. jejuni* on retail poultry may affect food safety [34]. 

It has been well documented that contaminated chicken is frequently involved in human campylobacteriosis [24]. The findings in this study demonstrate that duck meat is also an important source that can transmit *Campylobacter* to humans since *Campylobacter* was more frequently isolated from retail duck meat (62.3%) than chicken (50.4%). Similarly, Wei et al. have reported that *Campylobacter* contamination is more frequent on duck meats than raw chicken in Korea [35]. According to the study of Little et al., 50.7% of raw duck meat in the UK is contaminated with *Campylobacter*, which is lower than *Campylobacter* contamination in raw chicken (60.9%) but greater than that in turkey (33.7%) [36]. The frequencies of *Campylobacter* isolation from raw chicken were higher in summer (56.8%) than in winter (40.4%). Similarly, the prevalence of *Campylobacter*-positive broiler flocks is significantly higher in summer (54~60%) than during the rest of the year (14~48%) in the UK [37]. However, such a seasonal pattern of *Campylobacter* isolation from chicken was not observed in duck meat, as the isolation frequency of *Campylobacter* from duck meat was similar between summer (63.6%) and winter (60.7%). Although the reason for the different seasonality patterns in *Campylobacter* isolation remains unexplained, the results clearly showed that duck meat is more likely to be contaminated with *Campylobacter* regardless of the season. 

Duck meat isolates of *C. jejuni* harbored most of the tested virulence genes. However, the prevalence of the virulence genes was relatively lower in duck meat isolates compared to chicken isolates. Particularly, the genes involved in invasion (e.g., *cia*B) and toxin production (i.g., *cdt*B) were less prevalent in *C. jejuni* strains from duck meat compared to chicken isolates. The prevalence of *vir*B11 was low in both chicken and duck meat isolates. The prevalence of virulence genes was higher in *C. jejuni* strains from retail poultry in Korea compared to previous studies performed in other countries. The *iam* locus has been detected in 54.7% and 57.1% of *C. jejuni* isolates from chicken in Poland and Canada, respectively [21,38]. However, 97.8% and 88.9% of *C. jejuni* isolates from raw chicken and duck meats, respectively, were positive for *iam* in this study. Similarly, the *pld*A and *cia*B genes have been detected from *C. jejuni* isolates from chicken carcasses at the frequencies of 63.6% and 67.3%, respectively, in Brazil [39], 56% and 40% in the US [40], and 84.3% and 68.6% in Canada [21]. In this study, 94.4% and 91.1% of *C. jejuni* isolates from raw chicken and duck meats, respectively, harbored *pld*A, and the *cia*B gene was detected in 95.6% and 88.9% of *C. jejuni* strains from raw chicken and duck meats, respectively. In *C. jejuni* strains from raw chicken, the virulence genes were highly prevalent regardless of the aerotolerance level. Similar to a previous report [21], however, some virulence genes, such as *cad*F, *pld*A, *doc*A, and *peb*1, were more frequently detected in AT and HAT strains of *C. jejuni* from duck meats than OS strains. Since the virulence genes tested in this study play a critical role in the pathogenicity of *Campylobacter*, high prevalence of the virulence genes in *C. jejuni* isolates from duck meat suggest that these strains are potentially a food safety threat. 

*C. jejuni* strains resistant to clinically important antibiotics, particularly FQs, were highly prevalent in retail raw poultry. High prevalence of FQ-resistant *C. jejuni* on retail chicken in Korea has been reported in multiple studies. Han et al. has reported that 92.2% of *C. jejuni* isolates from raw chicken in Korea are resistant to FQs [41]. Multidrug-resistant *Campylobacter* is highly prevalent in domestic and imported retail chickens in Korea, and FQ resistance is significantly high (95%) in *Campylobacter* strains from retail chicken [42]. In this study, we observed that FQ resistance was even higher in duck meat isolates (97.8%) than chicken isolates (85.6%). Wei et al. also reported that *C. jejuni* strains from duck meat (87.8%) were more resistant to FQs than the strains from chicken carcasses (83.3%) [34]. FQ resistance in *C. jejuni* from ducks and retail duck meats substantially vary depending on the country. In Malaysia, 76% of *C. jejuni* isolates from ducks and their farming and processing environments were resistant to FQs [43]. Among nine *C. jejuni* and 11 *C. coli* strains isolated from duck meats in the UK, FQ resistance was detected in 54.6% of *C. coli*, and none of the *C. jejuni* strains were resistant to FQs [35]. Since FQs are clinically important for treating gastroenteritis, the high prevalence of FQ-resistant *C. jejuni* in duck meat may be a serious public health concern. However, resistance to macrolides was low in *C. jejuni* strains from retail poultry.

MLST CC-21 and CC-45 were dominant in duck meat isolates. Compared to chicken isolates, the sequence type of 28.9% of duck meat isolates was not determined by MLST; this suggests that different *C. jejuni* populations may exist on duck meat. A study from Korea has reported that CC-21 and CC-45 are the primary MLST sequence types of *C. jejuni* isolates from duck farms [44]. In New Zealand, CC-45 and CC-1034 were dominant CCs in *C. jejuni* from mallard ducks inhabiting the public access sites of urban areas [45]. Although only a limited number of studies are available regarding the MLST sequence types of *C. jejuni* strains from ducks, CC-45 appears to be a common MLST sequence type in *C. jejuni* strains from duck and duck meat. Approximately 45.5% of HAT *C. jejuni* isolates from duck meat belonged to CC-45, whereas most (66.7%) of the OS strains of *C. jejuni* from duck meat were not typable with MLST. 

By analyzing the prevalence of virulence genes and antibiotic resistance, we discovered that antibiotic-resistant and pathogenic *C. jejuni* populations, such as Types 4 and 13, are predominant on retail raw chicken and duck meats. Type 4 was more prevalent in HAT strains of *C. jejuni* compared to OS strains, whereas Type 13 was more dominant in OS strains than HAT strains. In addition, all nine strains belonging to MLST CC-443 were classified to Type 4, indicating that MLST CC-443 may be associated with antibiotic-resistant and pathogenic *C. jejuni* populations. *C. jejuni* strains belonging to Types 4 and 13 were predominant in summer (64.8%) compared to winter (40.5%). It has been reported that ambient temperatures are related to human infections with *Campylobacter* and human campylobacteriosis frequently occurs in summer [46]. *Campylobacter* is isolated from diarrheal patients in Korea more frequently in summer than in winter [47]. Along with the other factors affecting foodborne infections, such as cross-contamination and temperature abuse, the findings in our study suggest that *C. jejuni* strains with an increased public health risk of antibiotic resistance and virulence gene prevalence are more prevalent on retail raw chicken and duck meats in summer than in winter. 

## 5. Conclusions

This study performed an extensive comparative profiling of *C. jejuni* strains isolated from retail poultry (raw chicken and duck meat). *C. jejuni* populations with antibiotic resistance and virulence potential are highly prevalent on both retail raw chicken and duck meat. Since poultry is the major reservoir for *Campylobacter*, human exposure to *Campylobacter* is mainly associated with poultry sources. Since dietary patterns may vary in different countries and ethnic groups, foodborne exposure to *Campylobacter* may occur through the consumption of different kinds of poultry meat. Although some countries frequently consume duck meat, little is known about *C. jejuni* from duck. The findings in this study successfully demonstrated that *C. jejuni* strains from retail duck meat are highly antibiotic-resistant, aerotolerant, and potentially pathogenic. 

## Figures and Tables

**Figure 1 microorganisms-07-00433-f001:**
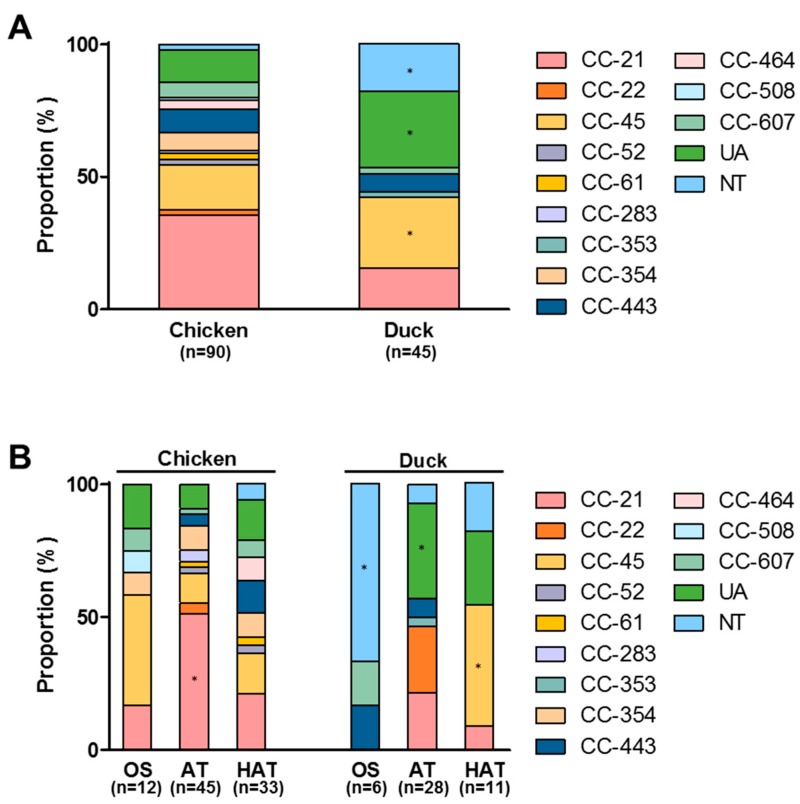
Multilocus sequence typing (MLST) sequence types of *C. jejuni* strains from retail raw chicken and duck meat (**A**), and the distribution of MLST clonal complexes (CCs) in *C. jejuni* strains with different aerotolerance levels from retail raw chicken and duck meat (**B**) in Korea. UA: unassigned to any CC defined, NT: not typable. *: *p* < 0.05.

**Figure 2 microorganisms-07-00433-f002:**
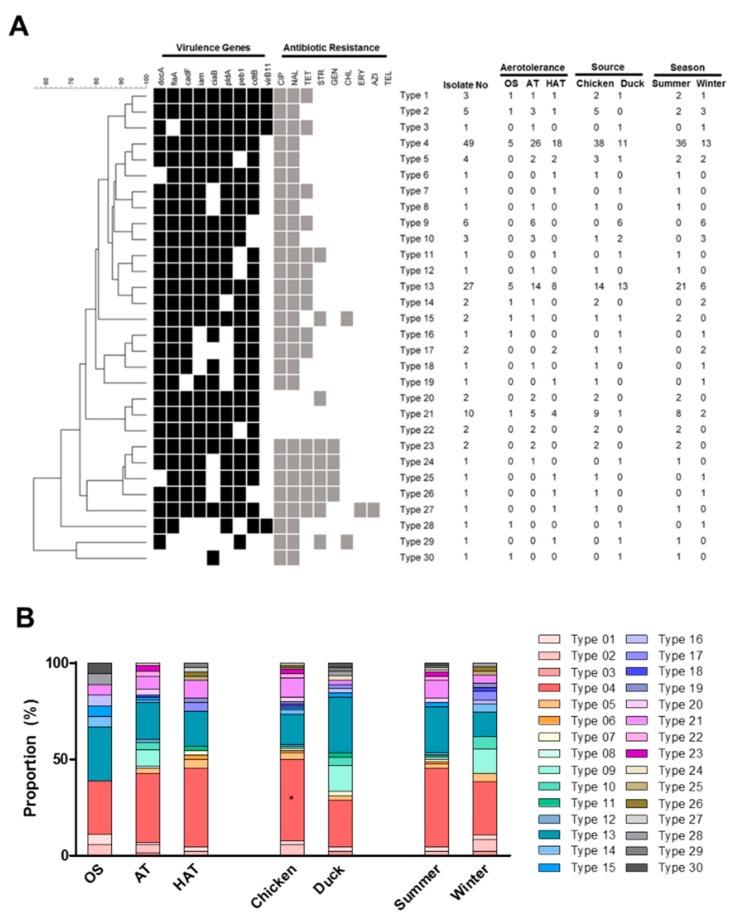
The virulence gene prevalence and antibiotic resistance (Viro and AMR) types of 135 *C. jejuni* isolates from retail chicken and duck meats in Korea (**A**) and the distribution of different Viro and AMR types depending on aerotolerance (OS: oxygen-sensitive, AT: aerotolerant, HAT: hyper-aerotolerant), origin (chicken or duck meats), and the season (summer or winter) (**B**). *: *p* < 0.05.

**Figure 3 microorganisms-07-00433-f003:**
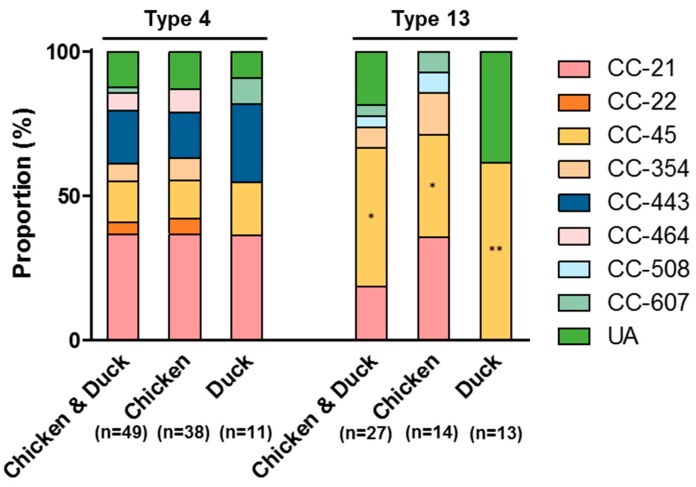
MLST sequence types of the two primary Viro and AMR types, Type 4 and Type 13, in *C. jejuni* isolates from retail raw chicken and duck meat. *: *p* < 0.05, **: *p* <0.001.

**Table 1 microorganisms-07-00433-t001:** Frequencies of *Campylobacter* isolation from retail raw chicken and duck meat.

Species	No. (%) of Retail Samples					
	Raw Chicken			Raw Duck Meats		
	Summer (*n* = 81)	Winter (*n* = 52)	Total (*n* = 133)	Summer (*n* = 33)	Winter (*n* = 28)	Total (*n* = 61)
*C. jejuni* only	26 (32.1)	12 (23.1)	38 (28.6)	10 (30.3)	9 (32.1)	19 (31.1)
*C. coli* only	10 (12.3)	6 (11.4)	16 (12.0)	5 (15.2)	3 (10.7)	8 (13.1)
*C. jejuni* and *C. coli*	10 (12.3)	3 (5.8)	13 (9.8)	6 (18.2)	5 (17.9)	11 (18.0)
Total	46 (56.8)	21 (40.4)	67 (50.4)	21 (63.6)	17 (60.7)	38 (62.3)

**Table 2 microorganisms-07-00433-t002:** Aerotolerance in *C. jejuni* strains isolated from retail raw chicken and duck meat.

Aerotolerance Level ^†^	No. (%) of *C. jejuni* Strains	
	Chicken (*n* = 90)	Duck (*n* = 45)
OS	12 (13.3)	6 (13.3)
AT	45 (50.0)	28 (62.2)
HAT	33 (36.7)	11 (24.4)

^†^ OS: oxygen-sensitive, AT: aerotolerant, HAT: hyper-aerotolerant.

**Table 3 microorganisms-07-00433-t003:** Antibiotic resistance of *C. jejuni* strains from raw chicken and duck meat.

Antibiotic	Breakpoint (µg/mL)	No. (%) of Resistant Strains	
		Chicken (*n* = 90)	Duck (*n* = 45)
Azithromycin	≥8	1 (1.1)	0 (0.0)
Erythromycin	≥32	1 (1.1)	0 (0.0)
Telithromycin	≥16	0 (0.0)	0 (0.0)
Chloramphenicol	≥32	1 (1.1)	2 (4.4)
Ciprofloxacin *	≥4	77 (85.6)	44 (97.8)
Nalidixic acid *	≥64	77 (85.6)	44 (97.8)
Gentamicin	≥8	4 (4.4)	1 (2.2)
Streptomycin	≥8	8 (8.9)	4 (8.9)
Tetracycline **	≥16	25 (27.8)	26 (57.8)

* *p* < 0.05; ** *p* < 0.001

**Table 4 microorganisms-07-00433-t004:** Prevalence of virulence genes in *C. jejuni* strains isolated from retail raw chicken and duck meat in Korea.

Virulence Gene	No. (%) of Strains	
	Chicken (*n* = 90)	Duck (*n* = 45)
*doc*A	88 (97.8)	44 (97.8)
*fla*A *	90 (100.0)	42 (93.3)
*cad*F	89 (98.9)	42 (93.3)
*iam* *	88 (97.8)	40 (88.9)
*cia*B	86 (95.6)	40 (88.9)
*pld*A	85 (94.4)	41 (91.1)
*peb*1	84 (93.3)	41 (91.1)
*cdt*B **	88 (97.8)	35 (77.8)
*vir*B11	7 (7.8)	3 (6.7)

* *p* < 0.05; ** *p* < 0.001

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
