# Peer review of "Comparative Analysis of Aerotolerance, Antibiotic Resistance, and Virulence Gene Prevalence in Campylobacter jejuni Isolates from Retail Raw Chicken and Duck Meat in South Korea"

_microorganisms, 2019, doi:10.3390/microorganisms7100433_

Round 1
Reviewer 1 Report
In general, this paper presents some interesting information, but the manuscript itself is not ready for publication. Some points should be clarified and a revision is needed.
Introduction
Page 2 lines 85-89: Only the aim of the study should be included. The results and conclusions should be included in the appropriate section.
Material and Methods
Page 2 lines 92-93: Details on chicken and duck meats analysed should be specified (whole carcasses, legs, wings,….).
Page 3 lines 94-100 the amount of poultry meat and enrichment broth should be indicated.
Page 3 lines 97 Why the enrichment broth was concentrated?
Page 3 line 105, MH supplier should be given.
Page 3 line 108. The centrifugation conditions should be described.
Results
Page 4, lines 149-150. This sentence should be eliminated, since it is included on material and methods section.
Page 4 lines 163-164: C jejuni was isolated in 81 poultry samples (53 chickens and 30 ducks. The authors selected 135 isolates (90 from chicken and 45 from duck). The criteria to select C.jejuni isolates should be described.
Discussion
Page 10, lines 340-348. Instead of a summary a conclusion according to the study carried out should be given.
Author Response
In general, this paper presents some interesting information, but the manuscript itself is not ready for publication. Some points should be clarified and a revision is needed.
Introduction
Page 2 lines 85-89: Only the aim of the study should be included. The results and conclusions should be included in the appropriate section.
Response: We corrected it according to the reviewer’s comment (page 2 lines 86-88).
Material and Methods
Page 2 lines 92-93: Details on chicken and duck meats analysed should be specified (whole carcasses, legs, wings,….)
Response: In the revised version, we specified the type of samples (Page 2 Lines 91, 94).
Page 3 lines 94-100 the amount of poultry meat and enrichment broth should be indicated.
Response: The whole carcasses were used for the enrichment in the experiment. We indicated this in the revised version (page 2 lines 93-100).
Page 3 lines 97 Why the enrichment broth was concentrated?
Response: Each poultry sample was enriched in 1 L of selective broth media. For the serial dilution and plating, only 100 µl can be used, which is a 10,000-fold dilution (100 µl/1 L). To improve the isolation efficiency, we first took 20 ml and concentrated for plating.
Page 3 line 105, MH supplier should be given.
Response: We indicated it in the revised version (Page 3, Line 108).
Page 3 line 108. The centrifugation conditions should be described.
Response: We described the centrifugation conditions in the revised version (Page 3, Lines 110-111).
Results
Page 4, lines 149-150. This sentence should be eliminated, since it is included on material and methods section.
Response: According to the comment, we deleted the sentence in the revised version (Page 4. Line 154).
Page 4 lines 163-164: C jejuni was isolated in 81 poultry samples (53 chickens and 30 ducks. The authors selected 135 isolates (90 from chicken and 45 from duck). The criteria to select C.jejuni isolates should be described.
Response: We explained it in the revised version. Due to the nature of the explanation, we added it to the Materials and Methods in the new version (page 3 lines 102-105).
Discussion
Page 10, lines 340-348. Instead of a summary a conclusion according to the study carried out should be given.
Response: We changed it in the revised version (Page 10 Line 345).
Thank you for the constructive comments.
Reviewer 2 Report
In this study, presence of C. jejuni in retail raw chicken and duck meat in Korea was investigated and the aerotolerance, MLST type, antibiotic resistance, and virulence gene distribution in a selection of strains were determined. There is a lot of work in this study, and results, particularly those related to duck are very interesting. However, I have some concerns regarding methodology that need to be addressed before being accepted for publication (see below). In particular, absence of any statistical analysis to support the comparisons made throughout the manuscript is a concern. Statements regarding comparison of results according to C. jejuni prevalence, aerotolerance, MLST type, resistance and virulence gene prevalence between the two sources tested (poultry vs. duck) or between seasons (summer vs. winter) are not supported by any statistical analyses. Therefore this is not a properly done comparative analysis but a simple description of the situation in both type of meat. Also, data presentation needs to be re-organized. In many parts of the manuscript, the text is a repetition of what is already shown in the Tables and Figures.
Some methodological aspects that need clarification:
Campylobacter isolation: According to M&M (L94-100), meat samples were subjected to enrichment, followed by concentration and resuspension in Bolton broth, which was then serially diluted and subcultured on supplemented Preston agar. What was the reason for concentration and resuspension? How many of those serial dilutions were plated on Preston agar? How many Preston agar plates were used per sample?
DNA preparation for PCR identification: The culture broth was resuspended in Muller Hinton (MH) broth to an optical density at 600 nm (OD600) of 0.1, diluted 10-fold in distilled water, boiled and cell debris pelleted by centrifugation and the supernatant used as template DNA. Which broth was resuspended in MH broth? The Bolton both? Then, identification was not carried out on single colonies?
L108, The isolates were further differentiated with 16S rRNA sequencing: I do not think that 16S rRNA sequencing allows differentiation between C. jejuni and C. coli.
PCR detection of virulence genes: was each gene analysed in a single PCR or did the authors perform any multiplexing? If so, describe.
I guess pure cultures from single colonies were used for MLST, MIC determination and PCR detection of virulence genes. In this case, how many isolates per sample were analysed by the mentioned techniques? I do not quite follow the numbers in the different parts of the manuscript:
- In Table 1, it is a bit confusing that the numbers in the last lane (Total) are not the result of adding up the three lines above (C. jejuni + C. coli + C. jejuni & C. coli). It took me a while to understand that the samples in line for C. jejuni & C. coli are already included twice as both C. jejuni and C. coli.
- How is it possible that 90 C. jejuni isolates from chicken and 45 from duck were tested for MLST typing, MIC determination and PCR detection of virulence genes if according to Table 1 C. jejuni was only detected in 51 chicken samples and 30 duck samples? Does this mean that the number of isolates tested per sample varies? If so, how did the inclusion of epidemiologically related isolates affect the results?
AST: Are the breakpoints indicated in Table 3 clinical breakpoints? I would think that epidemiological cut-offs are preferably used for food products. Data on antimicrobial resistance is only given at the level of individual antimicrobial agents, any interesting results on multidrug resistant profiles?
MLST: Why was the number of non-typable isolates so high? In L317, authors argue that it “suggests that different C. jejuni populations may exist on duck meat”. Again, does this mean that DNA used for typing was not isolated from pure cultures but was extracted from broth?
Statistical analyses: any statement regarding comparison of results (Campylobacter prevalence, aerotoloerance, MLST type prevalence, resistance and virulence gene prevalence) between the two sources tested (poultry vs. duck) or between seasons (summer vs. winter) needs to be supported by statistical analyses. At least a Chi-squared or Fisher test would be necessary when comparing each of the factors separately as it is done in many sections of the manuscript. But multivariate analysis such as logistic regressions would be more appropriate to take into account the possible effect of each of the variables tested in an integrative manner.
Minor changes:
Write the name of genes correctly throughout the manuscript, i.e., cadF rather than cadF (last letter should not appear in italics) L122: Define abbreviations OS, AT and HAT. Avoid using the word “interestingly” in the Results section. Do not refer to Tables and Figures in the Discussion (only in the Results section) Reorganize the order of the Supplementary Tables according to the order of citation in the text Fig S2 is not cited in the text L67: delete “during transmission to humans” L86: “compared” with what?Author Response
Campylobacter isolation: According to M&M (L94-100), meat samples were subjected to enrichment, followed by concentration and resuspension in Bolton broth, which was then serially diluted and subcultured on supplemented Preston agar. What was the reason for concentration and resuspension? How many of those serial dilutions were plated on Preston agar? How many Preston agar plates were used per sample?
Response: Each poultry sample was enriched in 1 L of selective broth media. Without concentration and resuspension, for a serial dilution and plating, only 100 µl can be used; this is a 10,000-fold dilution (100 µl/1 L). We used 20 ml of the enrichment broth for a serial dilution and plating to improve isolation efficiency (Page 3, Lines 96-99).
DNA preparation for PCR identification: The culture broth was resuspended in Muller Hinton (MH) broth to an optical density at 600 nm (OD600) of 0.1, diluted 10-fold in distilled water, boiled and cell debris pelleted by centrifugation and the supernatant used as template DNA. Which broth was resuspended in MH broth? The Bolton both? Then, identification was not carried out on single colonies?
L108, The isolates were further differentiated with 16S rRNA sequencing: I do not think that 16S rRNA sequencing allows differentiation between C. jejuni and C. coli.
Response: For the identification, we confirmed the genus using 16S sequencing and the species using the multiplex PCR. We explained it in the revised version (Page 3, Lines 102-107).
PCR detection of virulence genes: was each gene analysed in a single PCR or did the authors perform any multiplexing? If so, describe.
Response: Each virulence gene was detected using single PCR reaction (Page 3, Lines 128).
I guess pure cultures from single colonies were used for MLST, MIC determination and PCR detection of virulence genes. In this case, how many isolates per sample were analysed by the mentioned techniques? I do not quite follow the numbers in the different parts of the manuscript:
Response: Sorry for the confusion. Only one or two isolates were selected from each sample based on the morphological differences to avoid picking up identical clones. We described it in the revised version (Page 3, Lines 103-105).
- In Table 1, it is a bit confusing that the numbers in the last lane (Total) are not the result of adding up the three lines above (C. jejuni + C. coli + C. jejuni & C. coli). It took me a while to understand that the samples in line for C. jejuni & C. coli are already included twice as both C. jejuni and C. coli.
Response: Sorry for the confusion. We changed Table 1 in the revised version.
- How is it possible that 90 C. jejuni isolates from chicken and 45 from duck were tested for MLST typing, MIC determination and PCR detection of virulence genes if according to Table 1 C. jejuni was only detected in 51 chicken samples and 30 duck samples? Does this mean that the number of isolates tested per sample varies? If so, how did the inclusion of epidemiologically related isolates affect the results?
Response: In the revised version, we mentioned the number of isolates (please see our response above) (Page 3, Lines 103-105)..
AST: Are the breakpoints indicated in Table 3 clinical breakpoints? I would think that epidemiological cut-offs are preferably used for food products. Data on antimicrobial resistance is only given at the level of individual antimicrobial agents, any interesting results on multidrug resistant profiles?
Response: The CLSI provides clinical breakpoints of several antibiotics. We also referred to the breakpoints used by the FDA for the rest of the antibiotics, whose breakpoints were not indicated by the CLSI. We found this reference was missing in the previous version and included it in the revised version (Page 4, Lines 143, 144). Most strains were resistant to FQs and tetracycline. Thus, if the strains were resistant to other antibiotics, either erythromycin or gentamycin, they could be considered as multidrug resistant. Since the resistance profiles were straightforward, we did not mention multidrug resistance and rather emphasized the high prevalence of FQ resistance in the manuscript.
MLST: Why was the number of non-typable isolates so high? In L317, authors argue that it “suggests that different C. jejuni populations may exist on duck meat”. Again, does this mean that DNA used for typing was not isolated from pure cultures but was extracted from broth?
Response: Since the same DNA samples were used for PCR and 16S rRNA sequencing, we do not think it is because of the quality of DNA. Also, the portion of non-typable isolates was low in chicken isolates although we used the same methods.
Statistical analyses: any statement regarding comparison of results (Campylobacter prevalence, aerotoloerance, MLST type prevalence, resistance and virulence gene prevalence) between the two sources tested (poultry vs. duck) or between seasons (summer vs. winter) needs to be supported by statistical analyses. At least a Chi-squared or Fisher test would be necessary when comparing each of the factors separately as it is done in many sections of the manuscript. But multivariate analysis such as logistic regressions would be more appropriate to take into account the possible effect of each of the variables tested in an integrative manner.
Response: Based on the reviewer’s comment, we performed with a Chi-squared test and described statistical significance in the entire manuscript. Thank you.
Minor changes:
Write the name of genes correctly throughout the manuscript, i.e., cadF rather than cadF (last letter should not appear in italics)
Response: We corrected them in the entire manuscript
L122: Define abbreviations OS, AT and HAT.
Response: We defined them in the revised version (Page 3, Lines 125, 126)
Avoid using the word “interestingly” in the Results section.
Response: We deleted it in the entire manuscript.
Do not refer to Tables and Figures in the Discussion (only in the Results section)
Response: We deleted them in the revised version.
Reorganize the order of the Supplementary Tables according to the order of citation in the text Fig S2 is not cited in the text L67:
Response: We fixed the order in the text and the supplemental results. Thank you.
delete “during transmission to humans”
Response: We deleted it in the revised version (Page 2, Line 68).
L86: “compared” with what?
Response: We rephrased the sentence in the revised version (Page 2, Line 87).
Thank you for the constructive comments!
Round 2
Reviewer 2 Report
Most of my comments have been properly addressed by the authors.